# Evidence for an atomic chiral superfluid with topological excitations

Xiao-Qiong Wang[1,2,7], Guang-Quan Luo[1,2,7], Jin-Yu Liu[1,2], W. Vincent Liu[2,3,4,5 ✉], Andreas Hemmerich[6 ✉] & Zhi-Fang Xu[1,2 ✉]

Topological superfluidity is an important concept in electronic materials as well as ultracold atomic gases[1]. However, although progress has been made by hybridizing superconductors with topological substrates, the search for a material—natural or artificial—that intrinsically exhibits topological superfluidity has been ongoing since the discovery of the superfluid $^3$He-A phase[2]. Here we report evidence for a globally chiral atomic superfluid, induced by interaction-driven time-reversal symmetry breaking in the second Bloch band of an optical lattice with hexagonal boron nitride geometry. This realizes a long-lived Bose–Einstein condensate of $^{87}$Rb atoms beyond present limits to orbitally featureless scenarios in the lowest Bloch band. Time-of-flight and band mapping measurements reveal that the local phases and orbital rotations of atoms are spontaneously ordered into a vortex array, showing evidence of the emergence of global angular momentum across the entire lattice. A phenomenological effective model is used to capture the dynamics of Bogoliubov quasi-particle excitations above the ground state, which are shown to exhibit a topological band structure. The observed bosonic phase is expected to exhibit phenomena that are conceptually distinct from, but related to, the quantum anomalous Hall effect[3–7] in electronic condensed matter.

Quantum simulation involves the use of a comparably simple and precisely controlled synthetic quantum system to mimic poorly understood, isolated phenomena of a far more complex but less-controlled quantum system, while excluding the superimposed secondary structure that would impede a clear understanding[8]. Optical lattices—that is, ultracold neutral atoms trapped in laser-induced periodic potentials[9–11]—are well established systems that simulate an elementary class of many-body lattice model, the conventional $s$-band Hubbard model[12,13]. However, a large part of the physics that is relevant in electronic many-body systems is related to the coupling of electrons to magnetic fields via the Lorentz force, and hence remains inaccessible to quantum simulations in conventional optical lattices. Extensive research has been undertaken to establish a similar mechanism for neutral atoms in optical lattices, giving rise to so-called artificial magnetic fields or gauge fields[14,15] using dynamical techniques—similar to what is sometimes summarized as Floquet engineering for condensed-matter physics[16]. This has enabled the single-particle band structures of optical lattices to be endowed with built-in local[17] or even global[18,19] magnetic flux and the resultant single-particle topological properties; however, this comes at the cost of notable decoherence and markedly reduced lifetimes. An alternative approach towards optical lattice simulators beyond conventional $s$-band Hubbard physics is the implementation of orbital degrees of freedom by making use of higher Bloch bands[20–24]. This recent direction has led to interesting new multi-orbital quantum phases, including examples that have local angular momentum. An

intriguing, but as-yet unobtained, goal of this approach is to achieve a multi-orbital optical lattice quantum simulator that can, for example, capture bosonic versions of the topological superfluidity of paired electrons[25], or of quantum-Hall-related physics[26].

Here we take a step in this direction by demonstrating interaction-induced, spontaneous time-reversal symmetry (TRS) breaking and formation of a chiral atomic superfluid in an orbital optical lattice with global angular momentum and topological excitations and edge states. This metastable state exhibits remarkable robustness with a lifetime of hundreds of milliseconds. As an unequivocal signature of the prevalence of global angular momentum, characteristic momentum spectra are observed using time-of-flight spectroscopy. Our experimental observations directly show the spontaneous breaking of TRS in accordance with a mean-field tight-binding model and exact band calculations. The system is expected to exhibit the bosonic counterpart of the quantum anomalous Hall effect[3–7]; however, not as a result of an engineered band structure, as in the famous Haldane model[27]. Rather, as shown by theoretical considerations (Supplementary Information), it is the contact interaction between degenerate $p$ orbitals that leads to a spontaneous breaking of TRS, the formation of global angular momentum, a topologically non-trivial excitation band structure and the existence of topological edge states.

Our experiments use a boron nitride optical lattice that is composed of alternating shallow and deep potential wells—denoted A and B, respectively—arranged on a hexagonal lattice. As indicated in

[1]Department of Physics, Southern University of Science and Technology, Shenzhen, China. [2]Shenzhen Institute for Quantum Science and Engineering, Southern University of Science and Technology, Shenzhen, China. [3]Department of Physics and Astronomy, University of Pittsburgh, Pittsburgh, PA, USA. [4]Wilczek Quantum Center, School of Physics and Astronomy and T. D. Lee Institute, Shanghai Jiao Tong University, Shanghai, China. [5]Shanghai Research Center for Quantum Sciences, Shanghai, China. [6]Institute of Laser Physics, University of Hamburg, Hamburg, Germany. [7]These authors contributed equally: Xiao-Qiong Wang, Guang-Quan Luo. ✉e-mail: wvliu@pitt.edu; hemmerich@physnet.uni-hamburg.de; xuzf@sustech.edu.cn

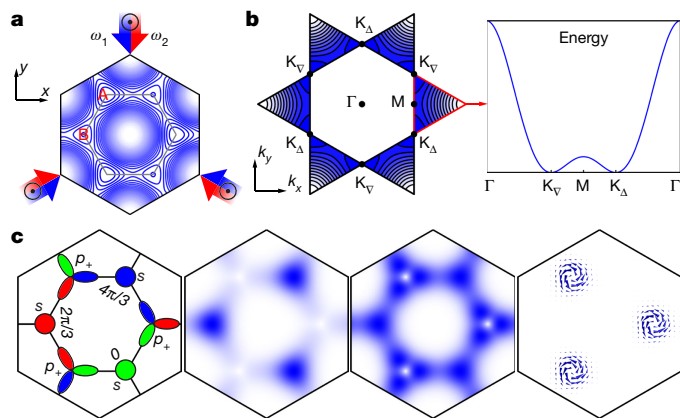

**Fig. 1 | Lattice set-up and single-particle band structure. a**, Lattice potential of the boron nitride optical lattice with two distinct local minima denoted A and B. The grey solid hexagon indicates the unit cell. **b**, Left, a contour plot of the second band of the boron nitride lattice across the second Brillouin zone. Right, the energy dispersion along the trajectory of high-symmetry points highlighted by the red triangle shows a double-well scenario. Two degenerate band minima arise at the $K_\Delta$ and $K_\nabla$ points. **c**, The first image shows the orbital composition of the second band Bloch function $\Phi_{K_\Delta}$ associated with the $K_\Delta$ point. The shallow wells host $s$ orbitals and the deep well hosts $p_+ = p_x + ip_y$ hybrid orbitals. The colours (green, red, blue) denote local phase values $v 2\pi/3$, $v \in \{0, 1, 2\}$. The region shown corresponds to the unit cell of $\Phi_{K_\Delta}$, which covers an area of three times the unit cell of the lattice potential. The second (third) panel illustrates the spatial distribution of $|\Phi_{K_\Delta}|$ when the $s$ orbitals in the shallow wells have lower (higher) energy than the $p$ orbitals in the deep wells. The fourth panel shows the mass current associated with the third panel.

Fig. 1a, the lattice potential is formed by three laser beams operating at a wavelength $\lambda$ of 1,064 nm. The beams propagate within the $x$–$y$ plane and intersect at an angle of 120°. Each beam comprises two spectral components at frequencies $\omega_1$ and $\omega_2$, with linear polarizations in the $z$-direction. Each set of spectral components with the same frequency forms a triangular lattice with adjustable amplitude. By tuning the frequency difference $\omega_1 - \omega_2$, the intensity patterns of both triangular lattices are shifted with respect to each other such that their sum produces a boron nitride lattice geometry with A and B wells. The relative potential offset of the two classes of well, denoted $\Delta V$, can be rapidly tuned. An optical dipole trap provides weak harmonic confinement with regard to the $z$-direction. For details, see Methods and Supplementary information.

Initially, a Bose–Einstein condensate of $^{87}$Rb atoms was loaded into the lowest Bloch band in the centre of the first Brillouin zone, denoted $\Gamma$ in Fig. 1b. The overall trap depth along the vertical $z$-axis—including the lattice potential, the dipole trap and gravitation—is 221 nK. As a key feature of our preparation protocol, we applied additional cooling by means of evaporation after the atoms were loaded into the lattice. The dipole trap depth decreases within 15 ms, such that the overall trap depth along the $z$-direction is reduced to 41 nK and the energetic atoms can escape. At this stage the atomic wavefunction is composed of $s$ orbitals residing in the deep B wells of the lattice. In a subsequent rapid quench the atoms are excited to the second band (compare with Fig. 1b) while remaining at the $\Gamma$ point, which constitutes a dynamically unstable energy maximum[28]. This is accomplished by tuning $\Delta V$ in approximately 100 μs according to the diagram in Fig. 2a, until the local $s$ orbitals in the B wells come to lie between the $s$ and $p$ orbitals in the A wells and hence belong to the second band. This also leads to a further reduction, to 24 nK, of the overall trap depth with respect to the $z$-direction, and thus an additional boost of evaporative cooling. Typical momentum spectra (Fig. 2c) recorded shortly (0.5 ms) after the quench show that a large number of Bloch states of the second band are populated with no apparent coherence[28]. Related excitation

protocols without additional evaporation have been used previously with bipartite square lattices[22,23] and with hexagonal lattices[29,30]. For details, see Methods and Supplementary Information.

The second band gives rise to a symmetric double-well scenario in quasi-momentum space. It possesses two inequivalent global energy minima located at the corners of the first Brillouin zone—denoted $K_\Delta$ and $K_\nabla$ in Fig. 1b—which exhibit perfect degeneracy, protected by TRS. A central observation of this work is that, in a re-condensation process, TRS is spontaneously broken and a condensate forms in either of the two K points with equal probability. This is shown in Fig. 2c, which displays momentum spectra recorded after variable holding times of between 0.5 ms and 755 ms. As time proceeds, sharp Bragg resonances grow at the $K_\Delta$ and $K_\nabla$ points. These resonances indicate the build-up of long-range coherence, and hence the formation of condensate fractions at $K_\Delta$ and $K_\nabla$. Notably, these condensate fractions—according to the observations—are generally not equally sized, as is shown in Fig. 2c, in which the three rows show examples with either a dominant $K_\Delta$ component (top), a dominant $K_\nabla$ component (bottom) or both components of similar size (middle).

We next point out that the observation of a dominant condensate fraction in either of the K points constitutes clear evidence of broken TRS. Exact band theory shows that the momentum spectra observed in the top and bottom rows of Fig. 2c are very well reproduced by the calculated momentum spectra of the Bloch functions associated with $K_\Delta$ and $K_\nabla$, respectively. For example, the calculated momentum spectrum of the Bloch function at $K_\Delta$ in Fig. 2b is plotted for comparison in the top row of Fig. 2c. Hence, the observation of population in a single K point unequivocally indicates the presence of a wavefunction that is well approximated by the Bloch function associated with that K point. Note that these Bloch functions, $\Phi_{K_\Delta}$ and $\Phi_{K_\nabla}$, inherently break TRS, as stated in Fig. 1c. The first panel schematically illustrates the composition of $\Phi_{K_\Delta}$ in terms of local $s$ orbitals in the shallow wells and $p_x$ and $p_y$ orbitals in the deep wells, which form $p_+ = p_x + ip_y$ hybrid orbitals ($p_-$ in case of $K_\nabla$). The three distinct colours indicate local phases that differ by $2\pi/3$, which results from an inherent vortical phase texture of $\Phi_{K_\Delta}$. In the second and third panels of Fig. 1c, $|\Phi_{K_\Delta}|$, which is derived from exact band theory, is plotted for two different settings of $\Delta V$, showing that different relative populations at A and B wells can be adjusted. The fourth panel shows the vortical mass current associated with $\Phi_{K_\Delta}$ for the same choice of $\Delta V$ as in the third panel. Note that the vortices residing at each deep well across the entire lattice all share the same sense of rotation—an interesting property that is associated with the threefold rotation symmetry of the hexagonal lattice in connection with the twofold degeneracy of the $p$ orbital manifold in two dimensions.

We define an experimental measure of chirality $\chi = (n_\Delta - n_\nabla)/(n_\Delta + n_\nabla)$ as the imbalance between $\Phi_{K_\Delta}$ and $\Phi_{K_\nabla}$ condensate populations, $n_\Delta$ and $n_\nabla$, respectively. This quantity is observed to be random in each experimental implementation with nearly zero average over many shots. However, although for short holding times—that is, in the initial phase of the condensation process—a distribution of $\chi$ with a maximum at zero is found, for long times beyond 100 ms this distribution develops pronounced maxima around ±0.5. This is shown in Fig. 3: in Fig. 3a, the average of the modulus of the chirality $\langle|\chi|\rangle$ is plotted against the holding time, whereas in Fig. 3b histograms of the distribution of $\chi$ are shown for holding times ranging from 35 ms to 505 ms. As is seen in Fig. 3a, $\langle|\chi|\rangle$ is small for short holding times of a few ms, and continuously increases until it reaches a maximum at around 0.4 for a holding time of about 265 ms, where the distribution of $\chi$ (compare with Fig. 3b) shows a pronounced bimodal structure. According to these observations, the condensation process develops in two stages: the first stage spontaneously breaks $U(1)$ symmetry, forming a condensate that is approximately described by an equal superposition of Bloch states $\Phi_{K_\Delta}$ and $\Phi_{K_\nabla}$, thus globally preserving TRS and zero net angular momentum. Alternatively, as in the one-dimensional double-well scenario of ref. [31], domains could be formed, such that locally TRS is broken but

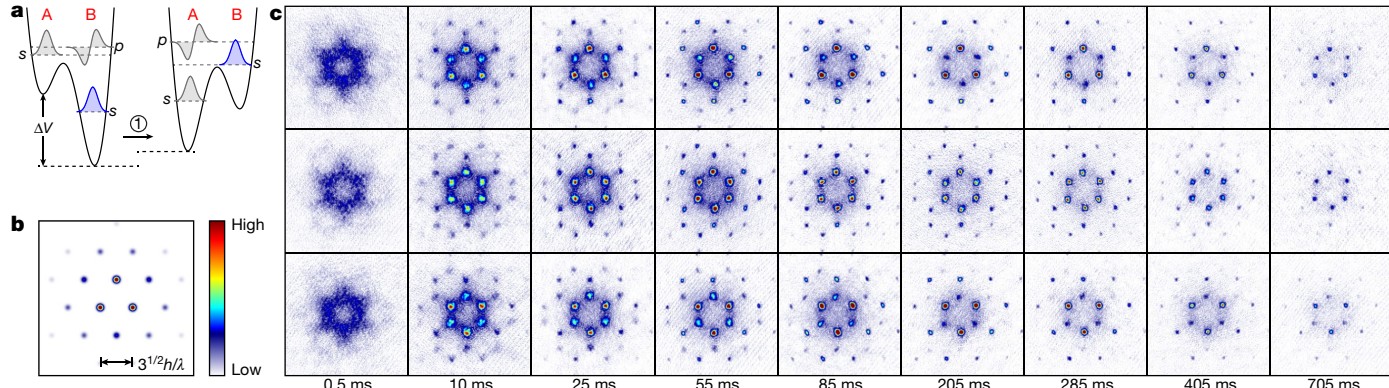

**Fig. 2 | Formation of second-band condensates. a**, The experimental sequence used to populate the second band. Initially, atoms are loaded into the s orbitals of the deep B wells (left). Subsequently, the relative potential offset of A and B wells is rapidly tuned until the local s orbitals in shallow B wells are centred between the s and the p orbitals in the deep A wells (right). For this setting, the s orbitals in the shallow B wells are predominantly populated and form the second band. **b**, Exact band calculation of the momentum distribution for the Bloch function $\Phi_{K_\Delta}$. $h$, Planck's constant. **c**, Momentum spectra at various holding times. The top and bottom rows correspond to second-band condensates with dominant condensate fractions at $K_\Delta$ and $K_\nabla$, respectively. The middle row shows condensates with nearly equal condensate fractions at $K_\Delta$ and $K_\nabla$.

the net angular momentum remains at zero. However, the narrow width of the observed Bragg peaks—seen in the momentum spectra with equally occupied K points in Fig. 2c—indicate well established coherence over large parts of the lattice, which is not compatible with a fine-grain domain structure. The second condensation stage breaks the global TRS by spontaneously bifurcating to form either of the two states $\Phi_{K_\Delta}$ or $\Phi_{K_\nabla}$, with equal probability.

Theoretical considerations (detailed in the Supplementary Information) suggest that the second stage of the condensation process is triggered by the ferromagnetic collisional interactions within the manifold of degenerate p orbitals in the deep lattice wells. In the presence of such collisions, the ground state according to mean-field theory is given by a condensate in either of the two K points rather than a superposition of both (see Supplementary Information). The atoms are self-organized into an order of synchronized vortices centred at each of the deep wells across the entire boron nitride lattice, as illustrated in Fig. 1c. The fraction of atoms populating the $p_+$ orbitals give rise to a non-zero global angular momentum. To access the ground state, which requires build-up of global orbital angular momentum, angular momentum must be exchanged with the environment. Indeed, a signature of this process is observed in the experiment. As shown in Fig. 3c, during the second condensation phase, past about 10 ms holding time, we observe a stream of atoms leaving the lattice perpendicular to the lattice plane (x–y plane) along the direction of gravity (−z-direction). This supports the interpretation that kinetic energy and angular momentum are transferred out of the lattice plane into the weakly confined z-direction, and finally escape from the lattice. It is of interest to note that, for stronger collisional interactions, under the assumption of quasi-momentum conservation, a condensate at the M points (Fig. 1b) is predicted to have a lower energy (Supplementary Fig. 5). However, a phase-separated mixed state with equal contributions from both K points might nevertheless be energetically more favourable.

The plots in Fig. 2c are obtained with $\Delta V$ adjusted during step 1 in Fig. 2a, such that most of the atoms in the second band reside in local s orbitals in the shallow wells and thus do not contribute notable orbital angular momentum. This corresponds to the wavefunction illustrated in the second panel of Fig. 1c. We now discuss how, by means of a slightly modified preparation protocol, a state with large global orbital angular momentum can be experimentally formed, which possesses a wavefunction according to the third panel of Fig. 1c. By adding an additional step (step 2 in Fig. 4a) 205 ms after the preparation protocol of step 1 in Fig. 2a, one may relocate a substantial portion of the atomic population towards the vortical p orbitals in the deep wells. This step

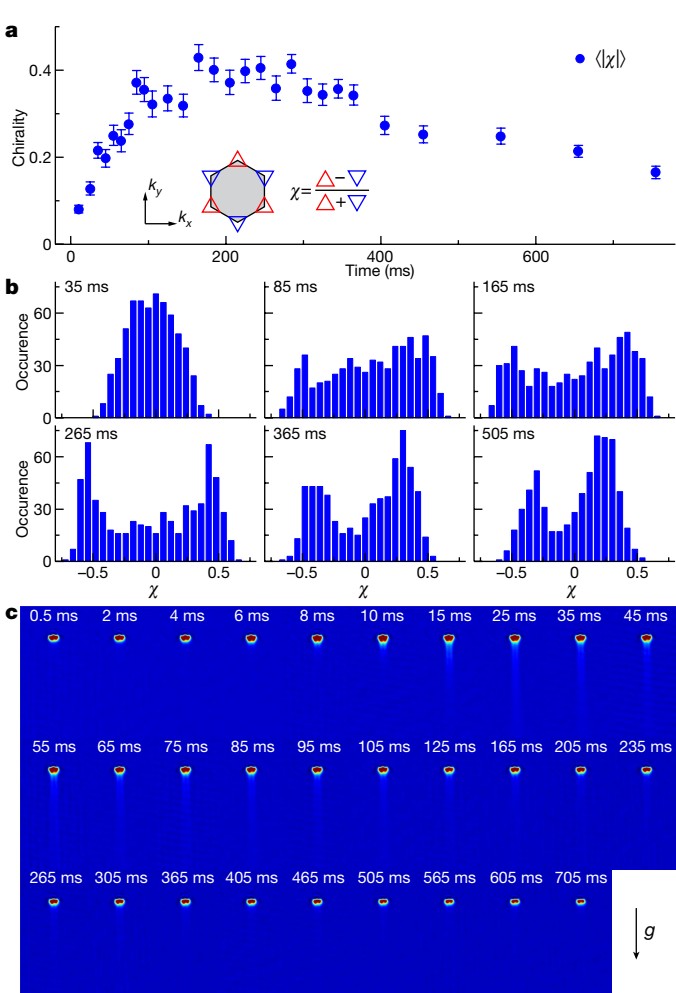

**Fig. 3 | Spontaneous breaking of TRS. a**, Evolution of chirality, which is characterized by the mean value of $|\chi|$. The inset specifies the definition of $\chi$. The error bars denote the s.e.m. for 46 experimental runs. **b**, Stacked histograms of $\chi$ for different holding times. **c**, In situ absorption images are recorded of a plane oriented perpendicular to the x–y plane for various holding times in the lattice. A stream of atoms escapes from the trap along the direction of gravity (−z axis) for holding times between ten and several hundred microseconds.

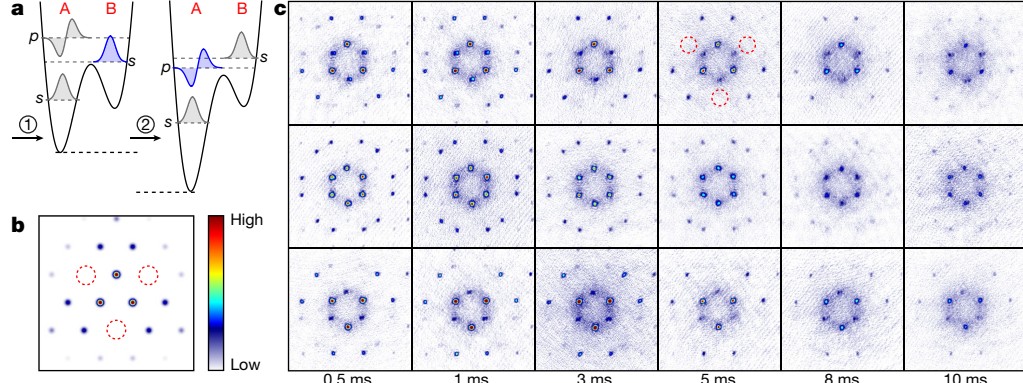

**Fig. 4 | Global angular momentum. a**, Illustration of the extended preparation protocol: after executing step 1 as in Fig. 2a, in step 2, $\Delta V$ is adiabatically tuned such that the atomic population in the second band is relocated towards the $p$ orbitals in the deep wells (see Methods and Supplementary Information). The absolute value of the corresponding wavefunction is shown in the third panel of Fig. 1c. **b**, Exact band calculations of the momentum

distribution for the Bloch function $\Phi_{K_\Delta}$ for dominant population of $p_+$ orbitals. The red dashed circles highlight the resulting absence of certain higher-order Bragg peaks, in contrast to Fig. 2b. **c**, Momentum spectra for extended preparation protocol at various holding times, analogous to those in Fig. 2c.

consists of adiabatically tuning $\Delta V$ (in 1 ms) until the local $s$ orbitals in the shallow wells lie energetically above the $p$ orbitals of the deep wells (see Methods and Supplementary Information). The absolute value of the corresponding wavefunction is shown in the third panel of Fig. 1c. In Fig. 4c we observe the resulting faster band relaxation that notably depletes the second band after only 10 ms. This observation indicates that the collision dynamics is now dominated by atoms in the $p$ orbitals of the deep wells, which provide faster decay due to increased overlap with the $s$ orbitals of the lowest band[23]. As a direct signature of dominant population of the $p$ orbitals, their additional nodes lead to destructive interference, thus preventing the occurrence of Bragg resonances within the areas enclosed by red dashed circles in Fig. 4c. This observation follows the exact band calculations in Fig. 4b. Note the contrast with the analogous spectra in Fig. 2b and with the corresponding experimental spectra in the top row of Fig. 2c. A large global orbital angular momentum arises, to which each atom in the $p_+$ orbitals contributes a portion ℏ. A quantitative analysis of the dependence of angular momentum on $\Delta V$ is shown in Supplementary Fig. 4.

Symmetric double-well scenarios in quasi-momentum space that lead to an interaction-induced spontaneous breaking of TRS and the formation of phases with staggered chiral order have been previously reported[22,31,32]. The long-lived atomic orbital superfluid devised in this work exhibits global angular momentum of the ground state wavefunction, although the Hamiltonian preserves TRS. As a consequence, basic Bogoliubov-de Gennes analysis verifies the emergence of bosonic topological excitations and edge modes (Supplementary Information). This opens up the prospect of studying these signatures experimentally. A direct comparison between experiment and theory would be enabled with regards to fundamental concepts that are related to, but have no previous analogue in, condensed-matter electronic and spin materials. Previous calculations have shown that, in a checkerboard square lattice with alternating shallow and deep wells and local non-zero angular momentum order, interaction-induced gaps should open in Bogoliubov quasiparticle spectra, and edge states protected by topological invariants should occur[33,34]. The proposed Zeeman-like bias is, however, an experimental challenge to implement for orbital degrees of freedom. Here, the boron nitride lattice resolves this challenge by spontaneously breaking TRS globally (driven by intrinsic interaction) with an emerging global angular momentum order. This paves the way to study dynamically controlled quasiparticles as exotic as a possible counterpart of Majorana fermions in a bosonic superfluid. The latter is widely sought-after in electronic topological superconductors, and the Supplementary Information presents calculations that support this expectation.

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

# Article

## Methods

### Realization of optical lattice

A two-dimensional hexagonal boron nitride (BN) optical lattice potential is created by three laser beams, propagating in the $x$–$y$ plane and intersecting at 120° angles. Each laser beam comprises two frequency components $\omega_1$ and $\omega_2$—which are both linearly polarized along the $z$ direction—with wavelength $\lambda \approx 1{,}064$ nm and therefore with negative detuning with respect to the relevant atomic transitions of rubidium atoms at 780 nm and 795 nm. The two frequency components are derived from two independent lasers. Experimentally, we first combine the two laser beams at frequencies $\omega_1$ and $\omega_2$, respectively, before splitting them into three beams. The total electric field for all beams is then written as

$$
\mathbf{E}(\mathbf{r}, t) = E_1 \mathbf{e}_z \sum_j \cos(\mathbf{k}_j \cdot \mathbf{r} - \omega_1 t + \theta_j)
$$
$$
+ E_2 \mathbf{e}_z \sum_j \cos(\mathbf{k}_j' \cdot \mathbf{r} - \omega_2 t + \theta_j').
$$

Here, the wave vectors are given by $\mathbf{k}_1 = k_{\mathrm{L}}(-\sqrt{3}/2, 1/2)$, $\mathbf{k}_2 = k_{\mathrm{L}}(\sqrt{3}/2, 1/2)$, $\mathbf{k}_3 = k_{\mathrm{L}}(0, -1)$, and $\mathbf{k}_j' = (\omega_2/\omega_1)\mathbf{k}_j$, where $k_{\mathrm{L}} = \omega_1/c$. $\theta_j = \omega_1 L_j/c + \theta_0$ and $\theta_j' = \omega_2 L_j/c + \theta_0'$, where $L_j$ denotes the optical path length of the $j$th beam from the splitting point to the centre of the lattice. The corresponding laser intensity $I(\mathbf{r})$ is proportional to the time averaging of the square of the electric field according to

$$
I(\mathbf{r}) \propto \frac{3}{2}E_1^2 + E_1^2 \sum_{\langle i,j \rangle} \cos[(\mathbf{k}_i - \mathbf{k}_j) \cdot \mathbf{r} + \theta_i - \theta_j]
$$
$$
+ \frac{3}{2}E_2^2 + E_2^2 \sum_{\langle i,j \rangle} \cos[(\mathbf{k}_i' - \mathbf{k}_j') \cdot \mathbf{r} + \theta_i' - \theta_j'],
$$

where the summation $\langle i,j \rangle$ is limited to $\langle 1,2 \rangle, \langle 2,3 \rangle, \langle 3,1 \rangle$. The generated optical lattice potential is proportional to the laser intensity and takes the form

$$
V_{\mathrm{BN}}(\mathbf{r}) = -V_1\left\{3 + 2\sum_{\langle i,j \rangle} \cos[(\mathbf{k}_i - \mathbf{k}_j) \cdot \mathbf{r} + (\theta_i - \theta_j)]\right\}
$$
$$
-V_2\left\{3 + 2\sum_{\langle i,j \rangle} \cos[(\mathbf{k}_i' - \mathbf{k}_j') \cdot \mathbf{r} + (\theta_i' - \theta_j')]\right\},
$$

where $V_{1,2} \geq 0$ for the relevant case of red detuning. Each of the two spectral components creates a triangular lattice potential, which sum up to form the total potential $V_{\mathrm{BN}}$. No interference terms arise because the frequency difference $\Delta\omega = \omega_1 - \omega_2$ is chosen in the range of a few GHz, which exceeds by far all relevant timescales of the atom dynamics. The relative position of the two triangular lattices is determined by the phase differences

$$
\Delta\theta_{ij} = (\theta_i - \theta_j) - (\theta_i' - \theta_j') = \frac{\omega_1 - \omega_2}{c}(L_i - L_j).
$$

Note that with $\Delta\omega \approx 2\pi \times 3$ GHz, a change of the lengths $L_i$ of the order of 10 μm corresponds to irrelevant changes of $\Delta\theta_{ij}$ of the order of $10^{-4} \times 2\pi$. Hence, $\Delta\theta_{ij}$ can be readily adjusted without the need for interferometric control of the lengths $L_i$. Experimentally, we choose convenient values for the lengths $L_1$, $L_2$ and $L_3$ and lock the frequency difference between the two lasers accordingly to fine-tune the relative position of the two triangular lattices appropriately to generate the desired boron nitride optical lattice. We set $(L_1 - L_2, L_2 - L_3, L_3 - L_1) = (-6.04, 3.02, 3.02)$ cm and $\Delta\omega = \omega_1 - \omega_2 = 2\pi \times 3.308$ GHz, which leads to $(\Delta\theta_{12}, \Delta\theta_{23}, \Delta\theta_{31}) = (-4\pi/3, 2\pi/3, 2\pi/3)$ and hence the boron nitride lattice potential

$$
V_{\mathrm{BN}}(\mathbf{r}) = -V_1\left\{3 + 2\sum_{\langle i,j \rangle} \cos\left[(\mathbf{k}_i - \mathbf{k}_j) \cdot \mathbf{r} - \frac{2\pi}{3}\right]\right\}
$$
$$
-V_2\left\{3 + 2\sum_{\langle i,j \rangle} \cos\left[(\mathbf{k}_i - \mathbf{k}_j) \cdot \mathbf{r} + \frac{2\pi}{3}\right]\right\}.
$$

The first (second) term describes the triangular lattice potential that gives rise to the A sites (B sites) in the combined boron nitride lattice shown in Fig. 1a. The potential difference between A wells and B wells can be readily adjusted on the microsecond timescale by tuning the ratio $V_1/V_2$. To avoid deformations of the lattice potential, the relative frequency difference between the two sets of triangular lattices and the laser intensities are carefully stabilized.

### Loading of lattice and detection schemes

A Bose–Einstein condensate of typical 40,000 ⁸⁷Rb atoms in the state $|F = 1, m_F = -1\rangle$ (where $F$ is the quantum number of total angular momentum and $m_F$ = magnetic quantum number) is prepared in an optical dipole trap formed by two crossed laser beams with trapping frequencies of $\{\omega_x, \omega_y, \omega_z\} = 2\pi \times \{26, 27, 71\}$ Hz. Including the gravitational force (pointing into the $-z$ direction) the trap depth along the $-z$ direction is 34 nK. A bias magnetic field of 1 G is applied along the $z$ axis. Within 120 ms, the lattice beam intensity is ramped up to $V_1 = 7.04 E_{\mathrm{R}}$ and $V_2 = 8.03 E_{\mathrm{R}}$, where $E_{\mathrm{R}} = h^2/2m\lambda^2$. At this stage the overall trap depth along the $-z$ direction is 221 nK. To provide additional evaporative cooling, after 5 ms, the depth of the optical dipole trap is ramped down in 15 ms, such that the overall trap depth along the $-z$ direction is reduced to 41 nK. Excitation into the second band is obtained by swapping the depths of the A and B sites via linearly changing $(V_1, V_2)$ to $(7.81, 7.23) E_{\mathrm{R}}$ rapidly in 0.1 ms. The trap depth along the $-z$ direction is thereby further reduced to 24 nK, which gives rise to further evaporation.

We recorded momentum spectra via time-of-flight spectroscopy or performed band mapping that enabled us to observe the quasi-momentum distribution. These techniques were used to derive the data in Fig. 2c. To obtain the data in Fig. 4c, the experimental protocol was slightly extended. After excitation to the second band and a subsequent holding time of 205 ms, the $p$ orbitals in the A wells are lowered by continuously increasing $V_1$ to $8.35 E_{\mathrm{R}}$ in 1 ms. Therefore, atoms are transferred from the $s$ orbitals in the shallow B wells to the $p$ orbitals in adjacent deep A wells. Momentum spectra of the atoms in the $x$–$y$ plane were obtained by switching off all potentials in less than 1 μs and, after a 20-ms-long ballistic expansion, performing absorption imaging. For band mapping, we decreased the intensity of the lattice exponentially with a time constant of 260 μs followed by 20 ms of ballistic expansion before performing absorption imaging. For the atom loss data shown in Fig. 3c, we performed in-situ absorption imaging of a plane perpendicular to the $x$–$y$ plane, after the atoms were excited to the second band and held there for a variable time.

### Data availability

All data presented in the figures are available upon request from the corresponding authors.

### Code availability

The numerical simulations were performed with MATLAB R2019b. Codes are available upon request from the corresponding authors.

**Acknowledgements** This work is supported by the National Key R&D Program of China (grant no. 2018YFA0307200), by the Key-Area Research and Development Program of Guangdong Province (grant no. 2019B030330001), by the NSFC (grant no. U1801661), by the high-level special funds from SUSTech (grant no. G02206401), by a fund from Guangdong province (grant no. 2019ZT08X324 (X.-Q.W., G.-Q.L., J.-Y.L. and Z.-F.X.)), by AFOSR grant no. FA9550-16-1-0006, by MURI-ARO grant no. W911NF-17-1-0323 through UC Santa Barbara, Shanghai

Municipal Science and Technology Major Project (grant no. 2019SHZDZX01) (W.V.L.) and by DFG-He2334/17-1 (A.H.).

**Author contributions** X.-Q.W. and J.-Y.L. carried out the experiments. G.-Q.L., Z.-F.X. and W.V.L. performed the calculations. Z.-F.X. devised the experiment with input by A.H. The lattice design was contributed by A.H. The manuscript was prepared by Z.-F.X, W.V.L. and A.H. All authors discussed the research.

**Funding** Open access funding provided by Universität Hamburg.

**Competing interests** The authors declare no competing interests.

**Additional information**
**Correspondence and requests for materials** should be addressed to W.V.L., A.H. or Z.-F.X.
