## [Peer Review File · Nature]

Manuscript Title: Evidence for an atomic chiral superfluid with topological excitations

Redactions – Mention of other journals

This document only contains reviewer comments, rebuttal and decision letters for versions considered at *Nature*. Mentions of the other journal have been redacted.

Reviewer Comments & Author Rebuttals

Reviewer Reports on the Initial Version:

Referees' comments:

Referee #1 (Remarks to the Author):

The manuscript entitled "Evidence for atomic topological superfluidity in a hexagonal optical lattice" describes an experimental study of Bose-Einstein condensation in the p-band of a hexagonal optical lattice. In contrast to the common condensation that takes place in the lowest-energy band, this p-band setting allows for interaction-induced time-reversal-symmetry breaking, and hence, for the potential creation of topological Bogoliubov excitations. In this sense, this optical-lattice system represents an interesting platform for the experimental study of topological superfluids.

More specifically, the authors demonstrate that atoms can be loaded into the higher ("p") band of their hexagonal optical lattice, using a quench protocol. Atoms are then shown to condense at two special points in momentum space, while preserving time-reversal symmetry. Finally, through a bifurcation (spontaneous time-reversal-symmetry breaking) process, the condensate selects one of these special points; this final state is associated with a finite angular momentum (as suggested by the theoretical analysis shown in Fig. 1c). During the last stage of condensation, a stream of atoms leaving the lattice is observed, which is compatible with an exchange of angular momentum with the environment [Fig. 3], and thus, with a build-up of angular momentum in the condensate. The authors also show how the condensation in the p-band can be optimized using an extended preparation scheme [Fig. 4].

In addition to these experimental results, the authors use a theoretical (Bogoliubov) analysis of their lattice model to argue that their p-band condensate should display topological properties (i.e. the calculated band structure exhibits non-zero Chern numbers and edge modes). This theoretical analysis, which is provided in the Supplementary (Section S6), builds on recent works by some of the authors [Phys. Rev. A 103, 013308 (2021); Phys. Rev. Lett. 117, 085301 (2016)].

I found this manuscript interesting and well written. In particular, I appreciated the analysis presented in Figs. 2-3, which nicely illustrates the different stages of the condensation in the higher band. However, I do not feel that these results are novel and important enough to justify publication in *Nature*, as I now explain:

(1) The loading of atoms into a p-band, followed by a "re-condensation" process, was previously observed in a similar setting using a similar protocol; see Hemmerich et al. *Nature Physics* 7, 147

(2011). Indications that such an orbital (p-band) superfluid can yield interaction-induced time-reversal-symmetry breaking were already presented in that previous publication.

(2) The title of the manuscript announces "Evidence for atomic topological superfluidity", however, this statement is misleading. Indeed, this "evidence" entirely relies on the theoretical analysis of the model (Section S6 in the Supplementary), which only suggests that this experimental setting *could potentially* exhibit properties of topological superfluidity; in other words, there is no experimental evidence at this stage. Based on the analysis of Section S6, it is not even clear how topological properties (such as Chern numbers and edge modes in the Bogoliubov spectrum) could be probed within this experimental setup; the very small gap in the Bogoliubov spectrum [Figs. S6-S7] even suggests that resolving these topological features will be extremely difficult in this context.

Altogether, my view is that this work reports on interesting advances in the field of Bose-Einstein condensation in higher bands. However, I don't feel that this progress is substantial enough to justify publication in Nature.

Referee #2 (Remarks to the Author):

This is a very exciting experiment. The researchers engineer a dimerized honeycomb optical lattice, and explore how a Bose gas responds to changing the potential. They interpret their results in terms of spontaneous time-reversal-symmetry breaking, and argue that the state they created has novel "topological" properties. This is clearly a topical subject which will attract significant attention.

It is useful to note that the model that they implement is similar to those that describe 2D layers of Boron-Nitride, or certain Moire structures. The connections to these material systems increases the impact of this work.

Given my excitement, I would be inclined to endorse publication in Nature, after the following questions/comments have been addressed to my satisfaction:

(1) The experiment has strong parallels with an experiment from the Chicago group (Ref. 32). The Chicago experiment also saw spontaneous symmetry breaking after a sudden change in a lattice. The geometry was different in the two cases, and the broken symmetries were different -- but the basic phenomenology looked very similar. A natural expectation (borne out by the Chicago experiment) is that this sort of quench experiment will lead to domains. I would like to see an explanation of why Wang et al. discount this possibility. The entire paper (especially the supplementary discussion) seems to assume that the cloud is homogeneous. To me the data is more easily interpreted in terms of a structure of domains.

(2) Figure 3a very nicely depicts the evolution of the total chirality: $\chi = (L-R)/(L+R)$ -- where L and R are the intensity of two different sets of Bragg peaks. If there are domains, then it would be useful (perhaps in the supplement) to separately plot L and R, rather than just the ratios. This would aid an interpretation in terms of coarsening and domain wall dynamics.

(3) It is clear that atomic motion in the z-direction is important for the dynamics (see for example Fig. 3c, and the discussion in the text). It would be very useful if the authors could estimate the trap depth in the z-direction. How does this compare to the bandwidth?

(4) The paper would benefit from further discussion of timescales and energy scales. For example, the authors should give an estimate of the band-width, mean-field energy scale, and collision time.

Otherwise it is hard to interpret statements about a 1ms ramp being "adiabatic" or 100 ms being "a long condensation time". An estimate of atom number and density would also be valuable.

(5) The data in Fig. 2 -- showing rapid decoherence followed by the emergence of order is reminiscent of the scenario in PRA 101, 033609 (2020). Some of the discussion from there may be relevant. Of course, the evaporation seen in Fig 3c means that there is active cooling in this experiment.

(6) It would be useful to see atom number vs time data.

(7) The motivator for the experiment shown in Fig. 4c is that the Bogoliubov spectrum in that case has a finite Chern number. I found the description of why this is important was a little cryptic, and filled with buzz-words. What is a "Majorana Boson"? How would the high energy topological edge modes manifest in an experiment?

(8) About 5 years ago there was an experiment by the Sengstock group [Phys. Rev. A 93, 033625 (2016)], which saw erroneous signatures of time reversal symmetry breaking in a similar geometry. It would be useful to verify that the same issues are not present here. [I don't believe they are -- but given the similarities, it seems essential to think about it.]

(9) I would recommend against using the acronym "QBCP". There is enough going on in the supplement that you shouldn't expect the reader to memorize arbitrary acronyms.

Author Rebuttals to Initial Comments:

Response to Reviewer 1

Reviewer 1 finds our work "interesting and well written" acknowledging that it "reports on interesting advances in the field of Bose-Einstein condensation in higher bands". He/she, however, expressed hesitation to endorse publication in Nature, proposing the following two reasons, both of which indicate to us, that we may not have sufficiently clearly communicated the far-reaching significance of our work and the fundamental differences to previous work. The two comments made by the reviewer have been a considerable help to strengthen our manuscript.

Reviewer 1 brings forward the following two criticisms:

Comment (1): The loading of atoms into a p-band, followed by a "re-condensation" process, was previously observed in a similar setting using a similar protocol; see Hemmerich et al. Nature Physics 7, 147 (2011). Indications that such an orbital (p-band) superfluid can yield interaction-induced time-reversal-symmetry breaking were already presented in that previous publication.

*Comment (2): The title of the manuscript announces "Evidence for atomic topological superfluidity", however, this statement is misleading. Indeed, this "evidence" entirely relies on the theoretical analysis of the model (Section S6 in the Supplementary), which only suggests that this experimental setting *could potentially* exhibit properties of topological superfluidity; in other words, there is no experimental evidence at this stage. Based on the analysis of Section S6, it is not even clear how topological properties (such as Chern numbers and edge modes in the Bogoliubov spectrum) could be probed within this experimental setup; the very small gap in the Bogoliubov spectrum [Figs. S6-S7] even suggests that resolving these topological features will be extremely difficult in this context.*

Response to comment (1):

The impression of the reviewer is correct that a re-condensation process, giving rise to time reversal symmetry breaking, has been previously observed in Nature Physics 7, 147 (2011). However, the quantum state achieved in the re-condensation process in the present work is fundamentally different. Also, the experimental protocol, that has enabled the present results, shows essential differences. We think the referee's reservation may be due to the circumstance that we have not sufficiently clearly explained the elementary differences with respect to research reported in Nature Physics 7, 147 (2011).

The topological chiral superfluid phase observed in the present experiment possesses *net angular momentum* order, breaking time-reversal symmetry *globally*. By contrast, the 2011 observed phase in the checkerboard lattice - and to be clear, all related states subsequently observed in the same lattice - possess a staggered order, which breaks the time-reversal symmetry *locally* (staggered from site to site), but not globally. In other words, the state discovered in Nature Physics 7, 147 (2011) is invariant under time reversal plus a translation by a primitive vector of the lattice, which is not the case in our present work. To put the remarkable significance of this subtle difference in comparison, the present phase and the state, previously explored in Nature Physics 7, 147 (2011), differ in terms of global symmetry and fundamental property in the same way as ferromagnetic and antiferromagnetic orders differ in spin systems. This difference is also the reason, why the Bogoliubov band structure is topological in our present work. Note that the observation of breaking of *global* TRS is possible in our system due to additional evaporation cooling in the lattice that provides exchange of atoms with the environment such that energy and quasi-momentum is not

conserved. Otherwise, one would expect domain formation, similarly as in the 1D double well scenario explored by the Chicago group in Nature Physics 9, 769 (2013).

Also, in methodological respect, our present work decisively differs from that in Nature Physics 7, 147 (2011). Unfortunately, we have practically missed to emphasize this point in the previous manuscript. In the present work, the condensation process itself is significantly refined by use of an optical dipole trap rather than a magnetic trap, which enables us to lower the trap depth along the z-direction after the atoms are loaded to the lattice potential, and thus add significant evaporative cooling to temperatures an order of magnitude lower than in the work previously reported in Nature Physics 7, 147 (2011). By choosing the hexagonal boron nitride lattice geometry, the chirality characteristic of the present superfluid is directly associated with the occupation of a single K-point in quasi-momentum space. The momentum spectra of the two possible K-points are unambiguously distinguishable and hence the formation of a single K-point state can be unequivocally detected. Note the fundamentally different situation in the Nature Physics 7, 147 (2011) experiment: There, the antiferromagnetic state is associated with the occupation of a superposition of two condensates at different high symmetry points (the X-points) with $\pi/2$ phase difference. Hence the detection of this state requires phase information and hence an interference set-up.

In the revised manuscript, we have made revisions to clearly emphasize the fundamental physical and methodological differences with regard to the research in Nature Physics 7, 147 (2011).

Response to comment (2):

We understand that the validity of evidence is indeed a rightful, important concern when evaluating experimental data and proposing physical interpretation. We would like to explain, why we think that the title of our work is not misleading. The reviewer's comment is general, not specific on a particular set of data we used as evidence. The concern, as it seems, is really about how much theoretical model or analysis one may use to extract the supporting evidence from analyzing experimental data. One could answer with Albert Einstein, who said: "It is the theory which decides what we can observe". We wish to take a less radical viewpoint by expressing our believe that it is fair to say, and that the reviewer can consent, that all advanced experiments rely on a certain level of theoretical modeling that allows one to design and interpret measurements. Definitely, one may expect that the theoretical modeling applied in this task should be well established and should not itself constitute a debatable open question. As we explain below, we firmly believe that the theoretical input used in our case belongs to the class of particularly well established pillars of condensed matter theory.

One of the well-known examples in the modern quantum physics is the report of observation of topological superconductors and non-Abelian Majorana fermions in condensed matter physics. Majorana fermions are quasiparticles, and they do not carry the same quantum number as electrons or holes, but a superposition of them. Measurements can only directly probe electrons. In condensed matter physics, the evidence for Majorana fermions (as edge states) has been mostly obtained from transport measurement, which relies heavily on BdG equations (for superconductivity) as the established theoretical model for quasiparticle excitations, etc. That in turn was broadly accepted as evidence - or proof - for topological superconductivity. Time-reversal symmetry breaking order parameter of Cooper pairing is another signature for topological superconducting phases such as p+ip order, but the complex nature of order parameters is often difficult to probe in electronic materials. Our case uses nearly the same level of theoretical modeling to draw the evidence as that in the case of topological superconductivity. For quasiparticles, we use the Bogoliubov analysis, a basic and

minimal theory that has been proved beyond doubt for its validity in Bose-Einstein condensates of bosonic atoms. The same applies to BdG equations for superconductors (for fermions). In atomic gases, the BEC order parameter is relatively easy to measure through the established momentum spectroscopy technique but transport is usually difficult to carry out in cold atom experiments. By contrast, the situation for electronic superconductors is reversed: to measure the time reversal symmetry breaking signature in the order parameter is difficult, while probing transport properties is comparatively easy. The cold atom and condensed matter experimental protocols are complementary; however, the level of the theoretical analysis to derive evidence from data is similar. In our case, momentum spectroscopy, in combination with the particular lattice geometry, enables us to unambiguously measure the order parameter and to observe the time reversal symmetry breaking signature in the course of its formation. Note that this is in contrast to the previous Nature Physics 7, 147 (2011) work, where the mere observation of momentum spectra does not allow one to determine the order parameter (See also our response to comment (1)). For concluding on the topological nature of the state formed in the present work, we employ the standard Bogoliubov analysis. This theory is so well established and elementary that this simply leaves no doubt about our conclusion. Transport measurements are much more difficult here, but their absence at present should not be taken as a lack of evidence. Our work will motivate an extended community of experimentalist to step in and explore new ways for complementary transport measurements.

We thank Reviewer 1 for her/his very useful specific comments that have helped us to better clarify the significance of our main results in the revised manuscript. We have added the following points:

- A. Highlight the additional evaporation cooling in the lattice that allows breaking of *global* TRS, formation of net angular momentum and topological excitation bands to occur in contrast to previous experiments, e.g. Nature Physics 7, 147 (2011).
- B. Highlight the two key features of the chosen lattice geometry that made this possible: the easy experimental distinguishability of single K-point condensates via their unique momentum spectra and the undeniable fact that such condensates exhibit broken TRS and topological excitation bands.
- C. In the Supplement, we included a brief comparison on the theory input required to probe electronic topological superconductors as compared to our case of a cold bosonic quantum gas. This should help to answer the concern expressed by Reviewer 1 in comment (2) and should be also helpful for a broader readership not familiar with the details of cold quantum gases.

In summary, the present manuscript reports a completely new interaction-driven topological superfluid phase with time-reversal symmetry breaking. Its difference from the p-orbital phase, reported in Nature Physics 7, 147 (2011), resembles the difference between ferromagnetic and antiferromagnetic order of spin systems. We used a previously established but significantly refined protocol to perform the experiment. The hexagonal lattice geometry and the application of extra evaporative cooling are crucial new ingredients to the qualitatively different result. We have made revisions in the manuscript, in order to emphasize this more clearly. We hope that with these remarks can help Reviewer 1 to appreciate the significance of our achievement and support publication in Nature.

Response to Reviewer 2:

We thank Reviewer 2 for her/his prospective endorsement for publication of our work in Nature after considering a number of well taken comments that we will respond to point by point below.

Comment #1:

The experiment has strong parallels with an experiment from the Chicago group (Ref. 32). The Chicago experiment also saw spontaneous symmetry breaking after a sudden change in a lattice. The geometry was different in the two cases, and the broken symmetries were different - but the basic phenomenology looked very similar. A natural expectation (borne out by the Chicago experiment) is that this sort of quench experiment will lead to domains. I would like to see an explanation of why Wang et al. discount this possibility. The entire paper (especially the supplementary discussion) seems to assume that the cloud is homogeneous. To me the data is more easily interpreted in terms of a structure of domains.

Response #1:

We agree that there are similarities between the phenomenology in our work and that of the Chicago group, although the underlying physics and its interpretation are quite different.

The Chicago experiment uses a one-dimensional lattice and applies driving to engineer the lowest single-particle Bloch band. Above a critical driving strength two degenerate local energy minima arise in quasi-momentum space for the lowest effective Bloch band. Two-body interaction favours occupation of a single minimum and hence the system is prone to spontaneous symmetry breaking. Both wells have opposite quasi-momenta and their occupation difference is addressed as magnetization. Since the total magnetization is conserved, in order to observe the ferromagnetic transition, a nonzero initial velocity of the atoms is introduced, acting as an external bias field. Otherwise, in the absence of an external bias field, domain formation is observed.

A double well scenario in quasi-momentum space (the two K-points in the second band) is also present in our experiment. Here, the two wells are associated with opposite angular momenta. Quasi-momentum is conserved and there is no bias field available. The energetic degeneracy of the two wells is protected by time-reversal symmetry of the Hamiltonian. Hence, we may expect either a condensate described by an equal superposition of Bloch states or domain formation as in the Chicago experiment. This definitely applies in the first stage of the condensation process, where both K-points are observed to be similarly occupied. However, the narrow width of the observed Bragg peaks, found in the momentum spectra with equally occupied K-points in Fig.2c, indicate well established coherence over large parts of the lattice, which is not compatible with a fine-grain domain structure. In the second stage of condensation, by evaporating atoms along the z-axis perpendicular to the lattice plane (Fig.3c), quasi-momentum is no longer conserved and a time-reversal symmetry broken condensate with a nonzero global orbital angular momentum is formed. Two key features of the hexagonal Boron Nitride lattice let us unambiguously demonstrate breaking of global TRS and the formation of global angular momentum: Firstly, the two K-points possess easily distinguishable momentum spectra such that the formation of a BEC that occupies a single K-point is unambiguously detected. Single K-point condensates are what we in fact observe in the long-time limit in our experiment. Secondly, the formation of a single K-point condensate is undeniably associated with global breaking of TRS and net angular momentum. This is fundamentally different from the lattice explored in Nature Physics 7, 147 (2011).

We thank Reviewer 2 for bringing up the comparison with the Chicago work and we have made revisions in the manuscript and Supplement to discuss this connection and the question of domain formation.

Comment #2:

Figure 3a very nicely depicts the evolution of the total chirality: $\chi = (L-R)/(L+R)$ - where L and R are the intensity of two different sets of Bragg peaks. If there are domains, then it would be useful (perhaps in the supplement) to separately plot L and R , rather than just the ratios. This would aid an interpretation in terms of coarsening and domain wall dynamics.

Response #2:

Actually, in Fig.3a, we show $\langle|\chi|\rangle$, the average of the modulus of $\chi = (L-R)/(L+R)$ with the brackets denoting averaging over many experimental runs. If we separately plot $\langle L/(L+R)\rangle$ and $\langle R/(L+R)\rangle$, the outcome is always close to $1/2$ and residual information on the degree of chirality may only show up in the standard deviations of the mean, which decrease with the number of averaged implementations. Below we show plots of $\langle L/(L+R)\rangle$ and $\langle R/(L+R)\rangle$ to illustrate this point.

In order to see domains, one could resort to the method used by the Chicago group, which however requires high resolution of the imaging optics to analyze time-of-flight spectra taken after short ballistic expansion times.

Comment #3:

It is clear that atomic motion in the z-direction is important for the dynamics (see for example Fig. 3c, and the discussion in the text). It would be very useful if the authors could estimate the trap depth in the z-direction. How does this compare to the bandwidth?

Response #3:

We thank the referee for this suggestion, which we consider very useful since, in fact, cooling via evaporation along the z-axis plays a crucial role here. We have made revisions in the Methods section and Supplement, in order to more clearly communicate this information to the readers. Here are the main steps of the evaporation protocol:

- (1) Before the lattice potential is activated, the dipole trap together with the gravitational potential provides a trap potential of 34 nK depth, where the lowest barrier against trap loss occurs in the z-direction, the direction of gravity.
- (2) After the lattice is turned on (with $V_1 = 7.04 E_R$, $V_2 = 8.03 E_R$), the effective total trap depth in the z-direction is about 221 nK.
- (3) For additional evaporative cooling, the depth of the optical dipole trap is ramped down in 15 ms, until a lower overall trap depth in the z-direction of 41 nK is realized.

(4) Finally, the lattice potential is rapidly changed to ($V_1 = 7.81 E_R, V_2 = 7.23 E_R$) in 0.1 ms for exciting the atoms into the second band. This further reduces the overall trap depth in the z-direction to 24 nK. The bandwidth of the second band is $0.1 E_R$, corresponding to about 10 nK.

Comment #4:

The paper would benefit from further discussion of timescales and energy scales. For example, the authors should give an estimate of the band-width, mean-field energy scale, and collision time. Otherwise it is hard to interpret statements about a 1 ms ramp being "adiabatic" or 100 ms being "a long condensation time". An estimate of atom number and density would also be valuable.

Response #4:

Following this suggestion, we provide the requested parameters in the revised manuscript. According to Fig.S3 of the supplement, the tight binding tunneling amplitude is about $0.29 E_R$. This corresponds to a tunneling time of about 1.7 ms. The duration of the excitation quench of 0.1 ms is hence significantly shorter than the tunneling time but also much longer than the onsite oscillation time scale on the order of $10 \mu\text{s}$, determined by the gap between the first and second bands. Furthermore, the peak density n_{peak} of the BEC in the dipole trap after evaporation cooling is about $4.7 * 10^{13} \text{ cm}^{-3}$. The mean-field interaction energy is $n_{\text{peak}} * 4\pi \hbar^2/m * (100.4 a_0) = \hbar * 365 \text{ Hz}$. We estimate the temperature as about $1/4$ of the trap depth of 34 nK, i.e. $T = 10 \text{ nK}$. The two-body collision parameter is $\beta = \sigma * v_{\text{mean}}$ with the scattering crosssection $\sigma = 8\pi * (100.4 a_0)^2$ and the mean velocity $v_{\text{mean}} = (16 k_B T / (\pi m))^{1/2}$. The associated collision time is $\tau = 1/(\beta n_{\text{peak}})$, which evaluates to $\tau = 14 \text{ ms}$. This is compatible with the observed condensation times on the order of several ten ms.

Comment #5:

The data in Fig. 2 - showing rapid decoherence followed by the emergence of order is reminiscent of the scenario in PRA 101, 033609 (2020). Some of the discussion from there may be relevant. Of course, the evaporation seen in Fig 3c means that there is active cooling in this experiment.

Response #5:

We thank the Reviewer for pointing out this reference. We agree that there is an initial dynamical instability leading to rapid decoherence before the re-condensation begins. A related situation in a one-dimensional lattice is explored in PRA 101, 033609 (2020). We thank the Reviewer to point this out to us. In the revised manuscript, we have added this paper as a new reference [29].

Comment #6:

It would be useful to see atom number vs time data.

Response #6:

The number of atoms condensed in the neighborhood of the K points versus time is shown in the figure below with the red solid line denoting an exponential fit with 295.6 ms decay time.

Comment #7:

The motivator for the experiment shown in Fig. 4c is that the Bogoliubov spectrum in that case has a finite Chern number. I found the description of why this is important was a little cryptic, and filled with buzz-words. What is a "Majorana Boson"? How would the high energy topological edge modes manifest in an experiment?

Response #7:

The reviewer's question is well taken. By "Majorana", we mean the quasi-particle operator is complex self-conjugate, such that the quasi-particle state is known as in real representation. We implicitly adopted this definition as standard from particle physics. In our Bose gas system, such real Majorana quasiparticles are made from a coherent superposition of bosonic particle and hole excited states with respect to the many-body condensate, i.e., a superposition of creation and annihilation atomic operators. In this manner, the term is actually a natural extension of "Majorana fermion", from electronic superconductors to the present atomic Bose gas. Yes, we should have added some definition of the term "Majorana boson" and explained a bit what we had meant.

We have decided to dispense the term "Majorana boson" to keep our paper self-contained and concise within the space limit. The respective sentence in the outlook of the main text has been revised to read: *This paves the way to study dynamically controlled quasiparticles as exotic as a possible counterpart of Majorana fermions in a bosonic superfluid.* The edge mode could be in principle detected by measuring the associated gap in the Bogoliubov spectrum with high resolution Bragg spectroscopy. Alternatively, a phase sensitive method could be used to measure a Berry phase or to map out the curvature of the lowest Bogoliubov band. We admit that such experiments are not easy. Our work provides a scenario, where such efforts would be greatly rewarded. We are not aware of any other scenario in cold gas physics providing the interesting perspective to explore interaction-induced topological Bogoliubov edge modes.

Comment #8:

About 5 years ago there was an experiment by the Sengstock group [Phys. Rev. A 93, 033625 (2016)], which saw erroneous signatures of time reversal symmetry breaking in a similar geometry. It would be useful to verify that the same issues are not present here. [I don't believe they are -- but given the similarities, it seems essential to think about it.]

Response #8:

In fact, in Phys. Rev. A 93, 033625 (2016), observations of time-reversal symmetry broken momentum spectra are discussed. This observation is not related to time reversal breaking in the underlying system. The reported signature of time-reversal symmetry breaking are

unequal populations among the first order Bragg peaks from different Brillouin zones belonging to the same quasi-momentum. The underlying mechanism is identified to be diffraction of the matter wave from the total density grating, which gives rise to a time-dependent interaction potential in the early stage of ballistic expansion. During ballistic expansion, the density grating can induce population transfer among momentum points representing the same quasi momentum. However, for our experiment, the two band minima for the second band are located at two different quasi momenta (K points) in the same Brillouin zone. The density grating during the early stage of ballistic expansion cannot induce population transfer among them, because the total quasi-momentum is conserved.

Comment #9:

I would recommend against using the acronym "QBCP". There is enough going on in the supplement that you shouldn't expect the reader to memorize arbitrary acronyms.

Response #9:

We follow the suggestion of the referee and explicitly write out the full term "quadratic band crossing point" instead of using the acronym "QBCP" in the supplementary material.

In summary, in contrast to previous experiments, the special design of our hexagonal boron nitride optical lattices enables us to implement additional stages of evaporative cooling after the atomic cloud is loaded into the lattice and into the 2nd band. Therefore, the atomic re-condensation process takes place, while the system is open and cooled. The "hot" particles are swept out in our experiment by lowering the dipole trap and allowing them to escape along the direction of gravity. These "hot" escaping atoms take away angular momentum, providing a crucial mechanism to enable the remaining atoms to set down to the lowest energy many-body ground state - a pure p+ip-orbital BEC with a net value of angular momentum. In other words, because the re-condensation takes place in a thermodynamically open environment, the system of condensed atoms is not subject to the constraint of particle number nor quasi-momentum conservation. Hence, domain formation, presumably governing the initial stage of re-condensation can be overcome as cooling proceeds.

List of major changes in main text:

Page 2, right column, 2nd paragraph

- Inserted discussion on the protocol for evaporative cooling in the lattice.
- Inserted new reference [29].

Page 3, right column, 2nd paragraph

- Inserted discussion on possible domain formation versus breaking of global TRS.

Page 5, left column, 1st paragraph

- Revised conclusion remarks

Page 6, right column, 1st paragraph

- Completed methods section with details on the evaporation protocol in the lattice.

List of major changes in Supplement:

Page 3, Sec.2

- Added detailed information on the protocol for evaporation cooling in the lattice.
- Added a paragraph, providing relevant time and energy scales.

Page 6

- Inserted revised figure S1.

Page 15

- Inserted a paragraph in Sec.6, where we discuss the role of Bogoliubov-de Gennes theory in the detection of topological superfluids/superconductors in electronic and quantum gas systems.

Reviewer Reports on the First Revision:

Referees' comments:

Referee #1 (Remarks to the Author):

I read the revised manuscript entitled "Evidence for atomic topological superfluidity in a hexagonal optical lattice", as well as the response file.

As I wrote in my previous report, I do appreciate the results presented in this work, and in particular, the experimental evidence that a TRS-breaking state can be achieved through successive condensation processes in p-bands. However, I am still not convinced that these experimental results support "evidence for topological superfluidity", in the sense that the topological nature of the superfluid is not manifest in the experimental data.

I agree with the authors that finding signatures of topological superfluidity is a subtle and complicated task in general; in fact, even edge-current measurements have been shown to be non-topological [PRB 91, 094507 (2015)]. However, the Bogoliubov analysis describing the topological band structure does suggest possible routes for detection. And the authors of this paper are totally aware of this: in PRL 117, 085301 (2016), they wrote "The topological features of the elementary excitations can be experimentally measured via coherently transferring a small portion of the condensate into an edge mode by stimulated Raman transitions" [the authors wrote a short Appendix on this scheme in PRL 2016]. Various proposals to detect topological features of the Bogoliubov band structure have been published in the last decade, both for 1D and 2D topological superfluids [PRB 83, 014513 (2011); NJP 14 113036 (2012); JPB 46 134005 (2013); Nature Comm. 5 4504 (2014); etc ...]. Observing "evidence of topological superfluidity", namely, demonstrating the topological nature of these exotic superfluids is extremely challenging; but once this will be achieved in the lab, this will undeniably deserve publication in Nature.

In my view, the current manuscript is an interesting follow-up paper to Nature Physics 7, 147 (2011), which will be of interest to the BEC community. I would therefore support publication in that same journal [redacted], provided that the authors change the title (in my view, the title should not suggest that the topological nature of the superfluid has been evidenced by the experimental data).

Referee #2 (Remarks to the Author):

The authors have adequately addressed my concerns, and I endorse publication in Nature. The data clearly shows that they have a condensate with spontaneously broken time reversal symmetry. This is an important observation.

My endorsement, however, is not unqualified. Referee 1 raised some valid objections, which I do not believe were adequately addressed: (1) The distinction between Ref 22 [Nature Physics 7, 147 (2011)] and the present manuscript is not directly explained in the text. I would recommend adding some discussion of this to the supplementary information. (2) The authors should seriously consider changing the title to more transparently convey the content of the paper. I also agree with Referee 1 that the authors have failed to explain the significance of the Chern number of the Bogoliubov spectrum. I could be wrong, but I suspect that there is very little relevance to that Chern number, and that it weakens the paper to focus too much on that feature.

Additionally, some copy editing is necessary to make the paper more accessible. The authors have a tendency towards long compound sentences which are hard to parse, and they continue to use more jargon than is strictly necessary. Furthermore, the paper would benefit from more clearly articulating the logic behind the experimental progression. For example, I don't think the main text ever explains why they do the experiment shown in Fig. 4.

Author Rebuttals to First Revision:

Response to Reviewers

Response to Reviewer 1

Reviewer comment 1.1

... I do appreciate the results presented in this work, and in particular, the experimental evidence that a TRS-breaking state can be achieved through successive condensation processes in p-bands.... However, I am still not convinced that these experimental results support "evidence for topological superfluidity", in the sense that the topological nature of the superfluid is not manifest in the experimental data. ... in my view, the title should not suggest that the topological nature of the superfluid has been evidenced by the experimental data

Our response 1.1:

We thank the reviewer for her/his cautious assessment of our use of the term "evidence". We in fact do not wish to promote the impression that the topological nature of the superfluid has been directly evidenced by the experimental data. However, the globally chiral state, which we have presented unequivocal evidence for, makes it practically inevitable to acknowledge the consequence of its interaction-induced topological character unless one is ready to dispense the generally accepted Bogoliubov-de Gennes excitation theory that is built on an experimentally observed mean-field ground state. To prevent any misinterpretation, we have changed our title to "Evidence for an atomic chiral superfluid with topological excitations". We believe that even though a direct observation of the topological character of the state prepared is not provided, the result of topological excitations supported by such a state does not require a new theory input but a standard, fully established framework to reveal it. We consider this property sufficiently important and reliable that it should be mentioned in the title. With the revised title, we see no risk of any misinterpretation.

Response to Reviewer 2

Reviewer comment 2.1

The distinction between Ref 22 [Nature Physics 7, 147 (2011)] and the present manuscript is not directly explained in the text. I would recommend adding some discussion of this to the supplementary information.

Our response 2.1

We agree that this is a very good idea, which interested readers might find quite helpful. We have added a detailed section (Sec.7) at the end of the supplementary information.

Reviewer comment 2.2

The authors should seriously consider changing the title to more transparently convey the content of the paper. I also agree with Referee 1 that the authors have failed to explain the significance of the Chern number of the Bogoliubov spectrum. I could be wrong, but I suspect that there is very little relevance to that Chern number, and that it weakens the paper to focus too much on that feature.

Our response 2.2

As discussed in the response to the according comment of Reviewer 1, we have changed our title to read "Evidence for an atomic chiral superfluid with topological excitations".

Reviewer comment 2.3

Additionally, some copy editing is necessary to make the paper more accessible. The authors have a tendency towards long compound sentences which are hard to parse, and they continue to use more jargon than is strictly necessary. Furthermore, the paper would benefit from more clearly articulating the logic behind the experimental progression. For example, I don't think the main text ever explains why they do the experiment shown in Fig. 4.

Our response 2.3

We have once more revised our manuscript to prevent unnecessarily long sentences and technical jargon. More importantly, we thank the Reviewer for her/his comment on "clearly articulating the logic behind the experimental progression". Actually, the main text in its previous form in the second paragraph of page 4 did explain that the experiment shown in Fig.4 aims to show that with a slightly modified preparation protocol, one can change the spontaneously formed state such that it possesses finite global angular momentum. In response to this comment, we have revised this paragraph to enhance its clarity. In addition, in the last paragraph of section S-4 of the supplementary information, a quantitative evaluation of the global angular momentum is found.

List of changes

- A. Title changed to read "Evidence for an atomic chiral superfluid with topological excitations" in response to Reviewer comments 1.1 and 2.2.
- B. Figures in main text replaced by high quality versions according to Nature style regulations.
- C. Page 4, right column: 2nd paragraph on physics of Fig.4 revised according to Reviewer comment 2.3.
- D. Supplementary, last two paragraphs on page 14: revised discussion on possible M-point condensation.
- E. Supplementary: New Section S-7 added in response to Reviewer comment 2.1.
- F. Supplementary: 4 new references added.
- G. Supplementary: Figures replaced by high quality versions according to Nature style regulations